# Long-Term Bovine Colostrum Supplementation in Football Players

**DOI:** 10.3390/nu15224779

**Published:** 2023-11-14

**Authors:** Mirosława Cieślicka, Błażej Stankiewicz, Radosław Muszkieta, Małgorzata Tafil-Klawe, Jacek Klawe, Anna Skarpańska-Stejnborn, Joanna Ostapiuk-Karolczuk

**Affiliations:** 1Department of Human Physiology, Nicolaus Copernicus University Ludwik Rydygier Collegium Medicum in Bydgoszcz, Karłowicza 24, 85-092 Bydgoszcz, Poland; m.cieslicka@cm.umk.pl (M.C.); malg@cm.umk.pl (M.T.-K.); 2Department of Physical Education, Kazimierz Wielki University in Bydgoszcz, 85-064 Bydgoszcz, Poland; blazej1975@interia.pl; 3Department of Physical Culture, Nicolaus Copernicus University, 87-100 Torun, Poland; muszkieta@umk.pl; 4Department of Hygiene, Epidemiology, Ergonomy and Postgraduate Education, Nicolaus Copernicus University Ludwik Rydygier Collegium Medicum in Bydgoszcz, M. Sklodowskiej-Curie 9, 85-094 Bydgoszcz, Poland; jklawe@cm.umk.pl; 5Department of Biological Sciences, Faculty of Physical Culture in Gorzow Wielkopolski, Poznan University of Physical Education, Estkowskiego 13, 66-400 Gorzów Wielkopolski, Poland; j.ostapiuk@awf-gorzow.edu.pl

**Keywords:** supplementation, bovine colostrum, immune effects, iron metabolism

## Abstract

Physical exercise, especially of high intensity, is a significant burden on an athlete’s body. It should be emphasized that achieving high results in competitive sports requires the use of significant, sometimes extreme, exercise loads during training, which may result in homeostasis disorders, adversely affecting the fitness of athletes. This study aims to investigate the effects of 6 months of bovine colostrum supplementation on indicators of immune system functioning, selected parameters related to iron management, and anabolic/catabolic balance in young football players. Twenty-eight male football players completed a double-blind, placebo-controlled crossover protocol (24 weeks of colostrum/placebo). A standardized exercise test was executed at the beginning of the trial and after 3 and 6 months of supplementation. Blood samples were taken before and after the exercise test and after 3 h of recovery. Markers of iron homeostasis, pro- and anti-inflammatory balance, and hormonal responses were determined. A significant increase in immunoglobulin G concentration was observed, accompanied by a decrease in inflammatory markers in supplemented athletes. Bovine colostrum supplementation had no significant effect on athletes’ performance or on iron management and hormonal response. The use of bovine colostrum, which is characterized by a high content of immunologically active compounds, can be an element of a relatively mild and safe intervention for reducing inflammation induced by intense physical exercise.

## 1. Introduction

Sports activity is associated not only with a significant physical load but also with exposure to extreme conditions, like psychological stress or environmental conditions, which determine the course and the strength of adaptive reactions. Football is an intermittent sport characterized by high-intensity physical exercise, which contributes to making it highly physiologically demanding. The repetitive and eccentric characteristics of football movements (accelerating, decelerating, kicking, jumping, and attacking) can induce muscle damage accompanied by a leakage of muscle enzymes and proteins into the circulation, oxidative stress, inflammation, and a decrease in muscle performance [1]. The characteristics of muscle damage after games can strongly disrupt the regeneration process. Given the busy schedule of elite players, consisting of two or even three games per week, their recovery periods do not allow for optimal restoration to pre-match levels, which may increase the risk of muscle injury [2].

Supplementation with bovine colostrum (BC) has a documented immunomodulatory effect on players’ diets and may be one of the elements of a safe and gentle intervention that restores homeostasis, i.e., the balance of the immune system. There is also increasing evidence that BC may be a valuable supplement for athletes to aid in exercise performance and recovery.

Bovine colostrum is the initial milk secreted by cows in the first days after delivery and contains chemical compounds with potential application in athletic performance improvement. Bovine colostrum includes, inter alia, growth factors (e.g., epidermal growth factor), antimicrobial peptides (e.g., lactoferrin), and immune mediators (e.g., immunoglobulins) that could be biologically active in humans because of their high concentrations and BC’s similarity to human colostrum [3]. Lactoferrin, an iron-binding glycoprotein, is involved in various physiological and protective actions, e.g., anti-inflammatory, antioxidant, antitumor, and antimicrobial activities. Superti [4] suggests that lactoferrin controls the inflammatory response by preventing iron-mediated free radical injury at inflamed sites through the control of oxidative stress; it modulates innate immune responsiveness, which alters the production of immune regulatory mediators. It has been proposed that, if physical exercise is intensive and/or prolonged, it can lead to a significant transient immune dysfunction (commonly referred to as immunodepression), which includes, inter alia, a reduction in cell- and mucosal-dependent parameters [5]. This brief period of immunosuppression after acute intense physical activity results in an immunological “open window”, wherein a sportsperson may be more susceptible to infection. This hypothetical period of dysfunction could influence the incidence of upper airway disease symptoms in people regularly performing prolonged exercise. Typical features of an exercise-induced immune response is an increase in the number of circulating leukocytes, mainly neutrophils, and an increase in the concentration of various substances that affect leukocyte function, including plasma inflammatory cytokines. The latter category includes tumor necrosis factor (TNF)-a; inflammatory macrophage-1 protein; interleukin (IL)-1β; and anti-inflammatory cytokines, such as IL-6, IL-10, and IL-1 receptor antagonist (IL-1ra) (Gleeson, M. Immune function in sport and exercise [6]).

Research conducted by Kostis et al. [7] on football players supplemented with a low dose of BC (3.2 g per day) showed a beneficial effect of the supplement that manifested, among others, in limiting damage to muscle fibers expressed by lower CK (creatine kinase) activity and in reducing inflammation (C-reactive protein and interleukin-6). Therefore, it can be assumed that extending BC supplementation to athletes to 6 months will alleviate inflammation by modulating the secretion of pro- and anti-inflammatory cytokines. Studies carried out so far have indicated that supplementation with BC may not only provide effective protection against the suppression of immunity in physically active people (especially competitive athletes, having high or even extreme training loads) but may also contribute to the stimulation of anti-inflammatory cytokine secretion and the suppression of pro-inflammatory cytokines. Bovine colostrum modulates cytokine production in human peripheral blood mononuclear cells stimulated with lipopolysaccharide and phytohemagglutinin [8]. Reducing post-exercise inflammation through the ingredients contained in BC has a positive effect on parameters related to iron metabolism [9]. It should be emphasized that disturbances in iron economy occurring during periods of increased training (competitions) are an important cause of decreased performance in athletes [10]. This study aims to investigate the effects of 6 months of bovine colostrum supplementation on indicators of immune system functioning, selected parameters related to iron management, and anabolic/catabolic balance in young football players.

## 2. Materials and Methods

### 2.1. Participants

The current study was a randomized clinical trial (No. ISRCTN13834604, date 7 November 2023) designed to compare the effects of 6 months of supplementation with bovine colostrum on the functioning of the immune system in football players. Participants were randomly assigned to study and placebo groups. A simple randomization method with computer-generated unique codes was used to assign the participants into the two groups (Figure 1). All subjects were informed about the purpose of the tests and procedures, and they voluntarily signed consent forms to participate in the experiment. The trial was not masked, but the staff who collected the research data outcomes were unaware of the study group assignments.

The study involved twenty-eight male athletes, football players playing in the III league Club, Chemik Moderator Bydgoszcz. The inclusion criteria were as follows: (1) competitive football training for at least 3 years; (2) male; (3) not taking any medications throughout the study; and (4) providing voluntary consent for participation in the study. The study participants did not report any health problems. Initially, twenty-eight volunteers were assessed as being eligible for the trial. The participants were assessed and randomly allocated to one of two groups: a supplemented group (*n* = 19) and a placebo group (*n* = 9). Table 1 presents the anthropometric data of the participants.

A flowchart of the recruitment process of the participants into this study is shown in Figure 1.

### 2.2. Physical Exercise Test

A multistage 20 m shuttle run test (Beep Test) [11] was conducted during the following training periods: in the preparatory period (T-1), at the beginning of the competitive period (T-2), and at the end (T-3) of the competitive period. The football players were informed about the test procedures and additionally motivated by a trainer to exert maximum effort. Each attempt was preceded by a warm-up, a 5 min low-intensity run (jog) (Figure 2).

The maximal multistage 20 m shuttle run test was controlled using a Lenovo ThinkPad X1 Yoga laptop computer, wired to Harbeth Super HL5 Plus speakers. The Beep Test was supervised by the team trainer (UEFA Pro Qualifications) using the program recommended by the Polish Football Association (PZPN) as a monitoring tool for examining changes in the level of cardio-respiratory fitness in all age categories of the football players.

### 2.3. Biochemical Evaluation

The effects of used supplements can be observed via serum biochemical parameters indicating iron homeostasis, inflammation processes, and hormonal balance. Thus, the biochemical parameters used in this study are the most suitable for showing these changes, and they can also be used for prognosis and evaluation of bovine colostrum supplementation in athletes.

During each exercise test performed in T-1, T-2, and T-3, blood samples were taken at three time points: before the exercise (Pre), post-exercise (Post), and after 3 h of rest (3 h restitution). Blood from the ulnar vein was collected in the amount of 2 × 9 mL to obtain serum for biochemical tests. The serum obtained from the competitors was frozen and stored for determinations at −80 °C. The results of the biochemical tests were determined using the following sets: IGF-1 (insulin-like growth factor 1)—determined using the DRG ELISA test; testosterone—determined using the DRG ELISA test; cortisol—determined using the DRG ELISA test; IL-10 (interleukin-10)—determined using the ELISA test DRG; IL-6 (interleukin-6)—determined using DRG ELISA; IgG—determined using SunRed ELISA; hepcidin—determined using DRG ELISA; TNF-α (tumor necrosis factor α)—determined using SunRed ELISA; and lactoferrin—determined using ASSAYPRO ELISA; iron + UIBC—determined via colorimetry using the colorimetric method (Bio-Maxima, Lublin, Poland). The unsaturated iron-binding capacity (UIBC) was calculated using the following formula: UIBC = TIBC–Fe. All parameters were determined using a spectrophotometer, SPECTROstar Nano (BMG LABTECH, Ortenberg, Germany).

### 2.4. Intervention—Diet Supplementation

Before the supplementation, the football players were randomly divided into two groups (Figure 1). The supplemented group (*n* = 19) received four gastro-resistant capsules of BC (produced by AGRAPAK, Poznań, Poland) every morning and evening. One gel capsule contained 0.4 g of colostrum. The composition of the supplement per daily dose of 3.2 g of colostrum (four capsules in the morning and four in the evening) was as follows: total protein—2.620 g; lactose—0.16 g; fat—0.05 g; and active protein substances (lactoferrin—30 mg; PRP (platelet-rich plasma)—0.16 g; IgG—1050 mg; IGF—16 µg; LZM—21.2 mg; and αLA—30 mg). The PRP content was estimated by measuring the content and ratio of amino acids (Pro and Val) based on the conducted research and analysis of bibliographic data [12]. The placebo group (*n* = 9) received powdered milk at the same dose, in the same form, and on the same dates as the supplemented group. The composition of the placebo was calculated for a single dose of 3.2 g: lactose, 1.6 g; protein, 1.08 g; fat, 0.04 g; and ash, 0.25. The supplementation period lasted for twenty-four weeks in total.

### 2.5. Statistical Analysis

A data analysis was performed using Statistica 13 (StatSoft, Cracow, Poland), and a graphical presentation of the results was developed using GraphPad Prism 8.4.0 (GraphPad Software Inc., La Jolla, CA, USA). The results of the variables are presented as mean ± standard deviation (SD). The level of significance was set at *p* < 0.05. The normality of the distributions was verified using the Shapiro–Wilk test. A paired sample *t*-test was performed to determine the differences between the results obtained at 3 and 6 months and the baseline for normally distributed data, and the Wilcoxon signed rank test was used for non-normally distributed data. To compare data at 3 time points (T-1: before, T-2: after 3, and T-3: after 6 months of supplementation), a repeated-measures analysis of variance (ANOVA) with Tukey post hoc test and Friedman’s test for non-normally distributed data were used. A two-way repeated-measures analysis of variance was used to confirm the interaction (group × time) and main effects. When a significant interaction (group x time) was observed, the Bonferroni post hoc test was conducted. For all measured variables, the estimated sphericity was verified according to Mauchly’s W test, and the Greenhouse–Geisser correction was used when necessary. Effect sizes were calculated using partial eta squared (η_2_ *p*), and <0.25, 0.26–0.63, and >0.63 were considered small, medium, and large effect sizes, respectively. Practical significance for pairwise comparisons was assessed by calculating Cohen’s d effect size. Effect sizes above 0.8, between 0.8 and 0.5, between 0.5 and 0.2, and lower than 0.2 were considered large, moderate, small, and trivial, respectively. The serum total-testosterone-to-total-cortisol ratio (T/C) was calculated as a proportion of the serum total testosterone concentration to the serum total cortisol concentration.

## 3. Results

According to Table 2, in both the supplemented and placebo groups, there were no significant differences in the results obtained during the exercise tests performed in the subsequent stages of the research. Moreover, no differences were observed between the study groups.

The exercise test performed in the T-1 period did not show any significant changes in the level of IgG in both the placebo and supplemented groups. After three months of supplementation, however, there was a significant increase in IgG in the supplemented group compared to the control (placebo) group at all time points (pre-exercise, post-exercise, and 3 h restitution) (*p* = 0.0033; η^2^ = 0.94 for the main effect of group), yet no significant effect of physical exercise on IgG levels was observed in both groups. The same situation was also observed after 6 months of supplementation (*p* < 0.001; η^2^ = 0.52 for the main effect of group) (Figure 3).

In both the supplemented and placebo groups, the intensive exercise tests performed in the T-1 period caused a significant post-exercise increase in IL-10, and this decreased after 3 h of recovery (*p <* 0.001; η^2^ = 0.46 for the main effect of time). In the T-2 period, a significant effect of exercise was visible only in the placebo group, where, after the exercise test, the levels of IL-10 significantly decreased during restitution compared to pre-exercise values (*p* = 0.0050; η^2^ = 0.25 for the main effect of time). In T-3, no significant changes were observed after the exercise test in both groups. The pre-exercise levels of IL-10 in both groups were significantly higher in the T-2 period than in the T-1 period. The post-exercise IL-10 value was significantly lower in the supplemented group in the T-3 period than in the T-1 period (*p <* 0.001; η^2^ = 0.30 for the main effect of time) (Figure 3).

In T-1, the level of IL-6 in the placebo group gradually increased after the exercise test and reached a significant difference after 3 h of recovery (*p =* 0.0035; η^2^ = 0.20 for the main effect of time). In the T-1 period, no significant differences were observed between the placebo and supplemented groups. In T-2, after the exercise test, a significant increase in IL-6 was observed in both the control and supplemented groups (*p <* 0.001; η^2^ = 0.60 for the main effect of time). After 3 h of recovery, the IL-6 level decreased below pre-exercise values. In T-3, the IL-6 level significantly increased after 3 h of recovery compared to pre-and post-exercise values in both groups (*p <* 0.001; η^2^ = 0.42 for group × time and *p <* 0.001; η^2^ = 0.71 for the main effects of time and *p <* 0.001; η^2^ = 0.71 for group). There were no significant differences between the placebo and supplemented groups. The pre-exercise and post-exercise values of IL-6 in T-2 were significantly higher than in T-1, and in T-3, the level of IL-6 was significantly higher than in the T-1 and T-2 periods (Figure 4).

There was no effect of intense exercise on TNF-α levels in T-1. In T-2, after the intense exercise test, a significantly higher level of TNF-α was observed in the placebo group after 3 h of recovery than in the control group. The changes in TNF-α levels were the greatest after 6 months of supplementation (T-3) (*p <* 0.001; η^2^ = 0.25 for group × time and *p <* 0.001; η^2^ = 0.47 for the main effects of time and *p <* 0.001; η^2^ = 0.62 for group). The exercise test caused an increase in TNF-α levels (significantly higher level post-exercise than pre-exercise) in both groups. However, after 3 h of recovery in the supplemented group, the level of TNF-α decreased, whereas in the placebo group, it remained increased, even after 3 h of recovery. Also, the pre-exercise TNF-α levels, as well as those after 3 h of recovery, were significantly higher in the placebo group than in the supplemented group (Figure 4).

In T-1, after the exercise test in the supplemented group, a significant increase in iron levels was observed after 3 h of rest. In the placebo group, an increase in the iron concentration was observed in post-exercise values and remained increased after 3 h of restitution (*p =* 0.0005; η^2^ = 0.25 for the main effect of time). In T-2, no significant differences in iron levels after the exercise test were observed in both groups. However, in the T-3 period, after the exercise test, an increase in iron levels in both groups was observed, and they decreased after 3 h of recovery in the supplemented group (*p* < 0.001; η^2^ = 0.31 for the main effect of time) (Figure 5). 

In the T-2 period only, a significant increase in hepcidin after exercise compared to pre-exercise values was observed in the control group (*p =* 0.0156; η^2^ = 0.15 for the main effect of time). In the remaining periods, a slight increase or no change at all in the response to exercise was observed. No differences were observed between the study groups or between the time points in subsequent periods (Figure 5).

The TIBC levels increased after the exercise test in each trial in both groups. During the period of restitution, the levels usually decreased to resting values (T-1: *p <* 0.001; η^2^ = 0.39; T-2: *p <* 0.001; η^2^ = 0.30; T-3: *p <* 0.001; η^2^ = 0.42 for the main effect of time). A significantly lower resting value was observed in T-3 than in T-2 in the supplemented group (Figure 6).

In T-1 and T-2, in both groups, the levels of UIBC increased significantly after exercise and decreased after 3 h of recovery (T-1: *p <* 0.001; η^2^ = 0.46, T-2: *p <* 0.001; η^2^ = 0.15 and T-3: *p* = 0.0472; η^2^ = 0.11 for the main effect of time). The pre-exercise value of UIBC was significantly higher in T-2 than in T-1 (Figure 6).

The exercise test performed in T-1 caused an increase in cortisol levels in the supplemented group after 3 h of recovery (*p <* 0.001 η^2^ = 0.38 for the main effect of time). The effect of exercise was also observed in T-3, where cortisol significantly decreased after the exercise test (*p <* 0.001; η^2^ = 0.32 for the main effect of time). A significantly higher level of cortisol, at all time points, was observed in the T-2 period than in the T-1 period. In T-3, the pre-exercise level of cortisol was significantly lower than in T-2 (Figure 7).

The testosterone levels after the exercise test performed in the T-1 and T-2 periods increased significantly compared to pre-exercise values during the 3 h of recovery (T-1: *p <* 0.001; η^2^ = 0.40; T-2: *p <* 0.001; η^2^ = 0.46; T-3: *p <* 0.001; η^2^ = 0.77 for the main effect of time). In the T-3 period, the resting testosterone values were significantly higher than in the T-1 period, and the physical exercise itself resulted in a significant decrease in the post-exercise period and an increase in the restitution period (Figure 7). There were no significant changes in the T/C ratio in both groups during the study.

In T-1, no significant changes were observed in lactoferrin levels after the exercise test. In T-2, in the supplemented group, the level of lactoferrin significantly decreased after exercise and remained at a significantly lower level during the restitution period (*p =* 0.0493; η^2^ = 0.10 for the main effect of time). In the placebo group, no significant changes were observed. In T-3, after the exercise test, the level of lactoferrin significantly decreased in both groups and then significantly increased after 3 h of recovery (*p <* 0.001; η^2^ = 0.60 for the main effect of time). Also, in T-3, a significantly higher level of pre-exercise lactoferrin in the supplemented group was observed than in T-1 (Figure 8).

In T-1, after the exercise test, a significantly higher level of IGF-1 was observed after 3 h of restitution in both groups. In T-2, a significant increase after exercise was observed only in the supplemented group after 3 h of rest (*p =* 0.0320; η^2^ = 0.12 for group × time and *p* < 0.001; η^2^ = 0.52 for the main effects of time). In T-3, no significant impact of exercise was observed in both groups. The pre-exercise values in T-2 in the supplemented group was significantly higher than in T-1, and, in T-3, the resting values were higher in both the supplemented and control groups than in T-1 (Figure 8).

## 4. Discussion

The use of bovine colostrum as a dietary supplement did not significantly improve the performance of the tested athletes (Table 2). Similar results were obtained by Kotsis et al. [7] in an experiment also conducted on football players who were supplemented with BC using the same dose as in our experiment (3.2 g/d). They showed a reduction in post-exercise muscle damage (EIMD) in the supplemented group, which, according to the authors, may also result in improved performance. In other studies [13], the supply of colostrum to physically active men (60 g/d for 8 weeks) resulted in an increased ability to exercise on a treadmill. The authors suggested that the obtained data indicated better post-exercise regeneration in the subjects. As mentioned before, bovine colostrum contains various nutrients that can support the regeneration of muscles and connective tissue after intense exercise. However, the scientific research on this topic is still limited, and it cannot be clearly stated that bovine colostrum significantly impacts athletic performance.

The use of enteric capsules containing colostrum on football players resulted in a significant increase in serum IgG levels compared to the placebo group (Figure 3). A higher level of this parameter was noted after 3 months of supplementation and remained at a similar level in the subsequent months of supplementation (the difference was also statistically significant concerning the placebo group). Similar results were obtained by Shing et al. [8], who showed that supplying cyclists with BC at a dose of 14.10 g/day prevented post-exercise IgG from lowering in the period of significant training loads. In other studies conducted on young basketball players (with the same dose and time of supply of the supplement), supplementation with BC did not cause statistically significant changes in the IgG concentration [14]. The main reason for these observed differences may be the variances in training loads. The supplementation of basketball players was conducted during a period with a significant reduction in training loads, while, for football players, the supplementation occurred in the competitive period, which is characterized by a significant increase in high-intensity exercises. Recently published studies on male endurance runners performing a 21 km maximal run showed statistically lower IgG concentrations compared to the resting value, at 2, 4, and 24 h after the end of the run [15]. Colostrum contains a variety of immune factors, such as antibodies (immunoglobulins, mainly IgG), lactoferrin, lysozyme, and defense peptides, involved in modulating the immune system. Increasing the concentration of IgG after supplementation with BC can strengthen the immune system, contributing to increasing its ability to fight infection [16]. This thesis is confirmed by a meta-analysis conducted by Jones et al. [17], who showed that oral BC supplementation reduces the frequency of upper respiratory symptom (URS) days and episodes in physically active people. Later studies by this team [18] showed that BC effectively reduces the decrease in response to a novel antigen during prolonged exercise, which may be a mechanism underlying the reduced number of URS reports. Other studies [19] showed that medical students supplemented with BC, who were potentially exposed to a higher risk of infection due to a significant workload and increased contact with infectious agents, had a reduced incidence of upper respiratory tract infections (URTIs).

Induced by intense or prolonged exercise, inflammation is a complex physiological and sometimes even pathophysiological body response. The data published so far suggest that inflammation can play a pivotal role in tissue repair and the elimination of pathogens, but, if left uncontrolled or not addressed in a timely manner, it can be harmful and negatively affect the entire body [20]. Research results indicate that the active compounds present in BC—with the ability to regulate the secretion of inflammatory mediators such as cytokines—can also affect a faster return to full training readiness [21]. Studies conducted on cell lines that were stimulated with lipopolysaccharide showed the inhibition of TNF, IL-6, and IL-4 release in the presence of BC and stimulated IL-2 secretion [22].

The exercise tests performed by the football players in three different periods of the training course (the preparatory period, and the initial and final parts of the competitive period) showed no differences in the level of IL-6 between the group supplemented with colostrum (BC) and the group receiving the placebo (Figure 3). It seems interesting that, at the beginning of the starting period (T-2), the players from both groups had significantly higher values of this parameter at rest and immediately after exercise. However, the discussed parameter acted differently in the next test period (T-3), where the increase in IL-6 occurred only after 3 h of rest.

The results of the presented studies indicate a reduction in the inflammatory response in the BC-supplemented group compared to the placebo group. The beneficial effect of the supplementation was visible in the T-2 and T-3 periods, which are characterized by relatively high training loads. In both periods, significantly lower post-exercise TNF-α values were observed in the supplemented group than in the control one. In addition, the longer 6-month supplementation in the athletes caused a quick return from the increased post-exercise level of TNF to baseline values, while, in the placebo group, the TNF-α level remained elevated, even after 3 h of restitution. 

Studies conducted on animal models with intestinal ischemia/reperfusion injury showed that the supply of colostrum to rats reduced the secretion of pro-inflammatory cytokines (IL-6, IL-1β, and TNF-α), with no differences between the groups in the level of anti-inflammatory cytokines (IL-10) [23]. In other studies [24], the effect of a 10-day BC supply was analyzed in athletes subjected to intense exercise loads. The authors showed no effect of BC supplementation on the analyzed parameters, while exercise caused a significant increase in IL-6, IL-10, and IL-1 receptor agonists and C-reactive protein, with no significant differences in other cytokines (interferon-γ, IL-1a, IL-8, or TNF-α). In our research, a post-exercise increase in IL-10 was shown only in T-1 (Figure 3), while, in the next study period (T-2), a decrease in this parameter was observed after 3 h of rest, but only in the placebo group. Similar results were obtained by Spagnuolo et al. [24], who analyzed the effect of the supply of bovine lactoferrin in rats subjected to intensive exercise. The research showed that the used supplement did not cause a significant increase in the concentration of IL-10 but contributed to a reduction in TNF-α concentrations. In vitro studies showed that bovine colostrum inhibits TNF-α production in human peripheral blood mononuclear cells collected from athletes and stimulated with lipopolysaccharide and phytohemagglutinin. According to the authors of the study, the above-mentioned process may indicate a shift in the balance towards an anti-inflammatory response [8]. An increase in TNF-α levels is associated with a transient increase in intracellular free or labile iron, which requires the activation of transcription factor NF-κB [25].

The adverse changes observed after physical exercise may be an effect of increasing hemolysis, which may not only intensify free radical processes (Fe ions catalyze the Fenton reaction) but may also increase the inflammatory response [26]. If post-exercise inflammation lasts for a long time, hypophoremia may develop, which, in turn, may lead to full-blown anemia [27]. The main mechanism underlying these processes is the retention of iron in the cells of the reticuloendothelial system, making iron unavailable for erythropoiesis. This reticuloendothelial sequestration of iron is mediated by hepcidin, which, by binding to ferroportin, is responsible for blocking iron transport from cells to plasma. Hepcidin transcription is a process strictly regulated by signaling molecules, including IL-6 [28]. As already mentioned, at the beginning of the competition period (T-2), a post-exercise increase in IL-6 was demonstrated in both groups of athletes (Figure 3). In contrast, the level of hepcidin, a hormone that regulates plasma iron levels, was significantly increased only in the placebo group (Figure 4). It seems that the key role in this process may be played by the lactoferrin present in BC, whose properties of iron ion chelation were described as early as the 1970s [29]. The beginning of the competitive period (T-2) also caused a decrease in the level of lactoferrin in the serum of the examined footballers, but only in the group supplemented with BC, with no significant changes in the placebo group (Figure 5). However, in the final stage of the research (T-3), in both groups after exercise, a decrease and then an increase in the parameter in question were shown. There were no differences between the supplemented group and the placebo group in any of the analyzed parameters of iron metabolism (Figure 4).

Only several studies have focused on BC supplementation and changes in cortisol and testosterone levels—hormones that indicate anabolic/catabolic balance. The cortisol response to physical effort observed in this study indicates unchanged or slightly decreased levels of cortisol in response to intense exercise in both the supplemented and placebo groups. 

These results may indicate that, in trained footballers, the test used caused a rather moderate stress reaction that tends to lower cortisol levels. However, the significant increase in cortisol in the T-2 period, compared to both the T-1 and T-3 periods, in the placebo and supplemented groups clearly indicates an increasing stress and fatigue associated with high training loads during the season. Similar results were observed not only in football players but also in football referees [30]. Moreover, Engelmann et al. [31] showed that, after reducing the training loads of footballers, the level of cortisol decreased, which was also observed in our research, wherein the level of cortisol decreased in the T-3 period, at the end of the competitive period, when the training loads were lower than those in the T-2 period. Since the response patterns were similar in the supplemented and control groups, it is difficult to indicate a positive or negative effect of the supplement administered. A lack of effect of BC supplementation on changes in cortisol levels during training was also observed by Mero et al. [32].

The changes in testosterone levels show that, during the T-1 and T-2 periods, the response pattern is also characteristic of the effect of intense exercise when testosterone increases after an exercise test. A significant increase in testosterone levels is indicated not only after strength exercises [33] but also after endurance exercises [34]. However, our study also showed an increase in the resting values of testosterone in the T-3 period. According to Rowell et al. [35], twenty-eight days of football training may increase testosterone levels by up to 18%. While an increase in resting testosterone levels may be an adaptive response to long-term training, a decrease in levels after intense exercise is rather unusual. According to Riachy et al. [36], the time point at which serum testosterone is measured is a strong factor that may affect the level of testosterone; in the case of our study, this may be the reason for the unexpectedly low level of testosterone observed after exercise in T-3. Also, in the case of testosterone, the response does not indicate the benefits of BC supplementation.

The long-term supplementation of football players with enteric capsules containing BC contributed to similar changes in this parameter in both groups (Figure 5). In the second study period (T-2) only, higher resting values of this parameter and a post-exercise increase in circulating IGF-1 were found in the supplemented group, without statistically significant differences to the placebo group. Similar results were also obtained in the works of other authors [37,38], which may confirm the safety of using BC as a dietary supplement in athletes. 

However, further research is necessary not only for the standardization of supplements containing colostrum in terms of the content of active ingredients but also for determining the optimal dose for physically active people.

## 5. Conclusions

There was no direct impact of bovine colostrum supplementation on exercise parameter improvement in athletes. However, it seems that the active ingredients contained in BC may indirectly improve the exercise capacity of players. The conducted research shows that supplying athletes with BC reduced inflammation, indicated by a significant decrease in TNFα secretion, which may result in faster muscle regeneration after exercise and, thus, may be reflected in better results in subsequent training sessions. However, understanding the in-depth mechanisms of colostrum action in this regard will require more detailed research.

In turn, the strengthening of the immune system, indicated by an increase in IgG in the subjects who consumed BC, may be an important factor in providing protection against infections, constituting an important element in maintaining the continuity of training by, e.g., reducing absences from training caused by infections. Therefore, it is necessary to consider supplying athletes with BC in periods of increased susceptibility to infections related to both the season (late autumn–winter–early spring) and periods with a high exercise load (competition period). It is worth noting, however, that the effectiveness of colostrum as a supplement to strengthen the immune system of athletes may vary depending on the quality and composition of each particular product, the time and dose of the supplement, and the training loads used.

## Figures and Tables

**Figure 1 nutrients-15-04779-f001:**
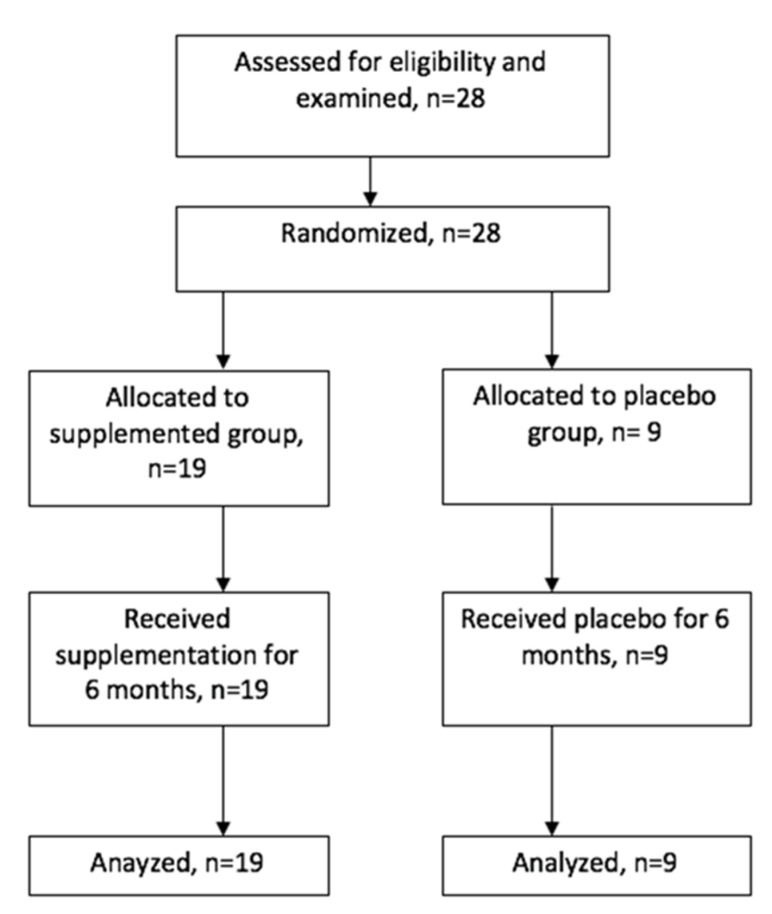
Recruitment process of participants into the trial.

**Figure 2 nutrients-15-04779-f002:**
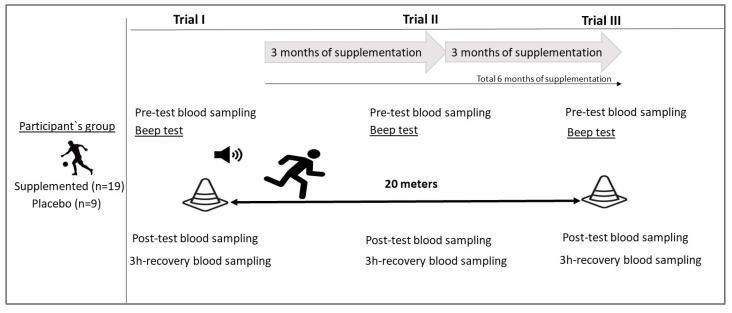
Study design.

**Figure 3 nutrients-15-04779-f003:**
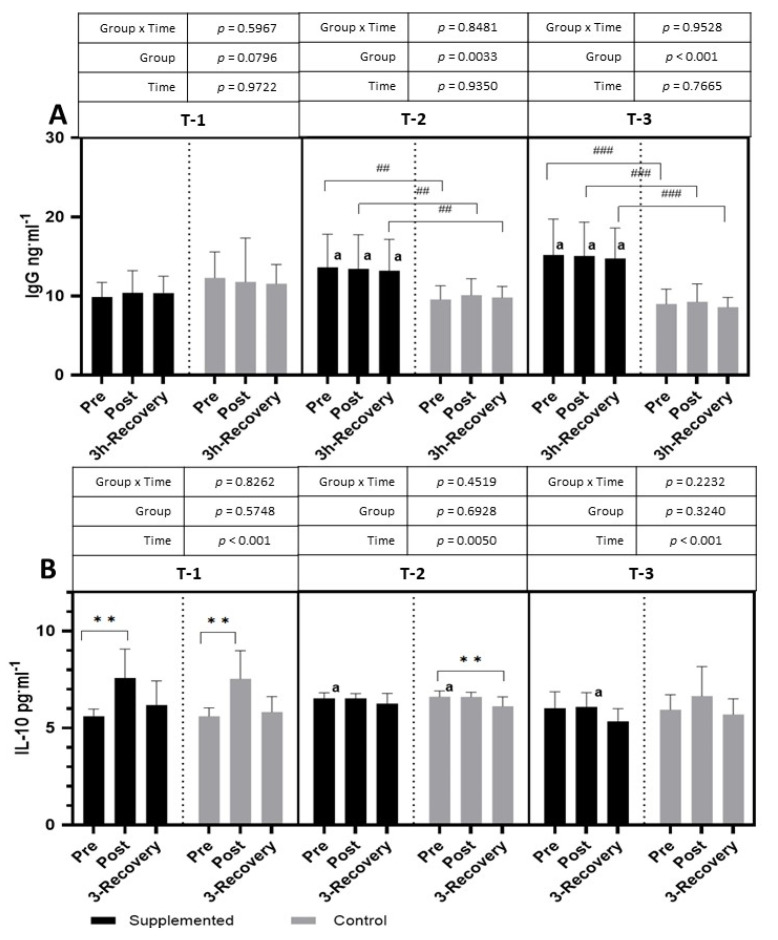
Effects of exercise and bovine colostrum supplementation on IgG (immunoglobulin G) (**A**) and IL-10 (interleukin-10) (**B**) in athletes. T-1: before supplementation (basal value); T-2: after 3 months of supplementation; T-3: after 6 months of supplementation. Data are means and standard deviation. ** *p* < 0.01, significantly different from pre-exercise in trials; ^##^
*p* < 0.01, ^###^ *p* < 0.001 significantly different between placebo (control) and supplemented groups in trials; ^a^ significantly different from T-1.

**Figure 4 nutrients-15-04779-f004:**
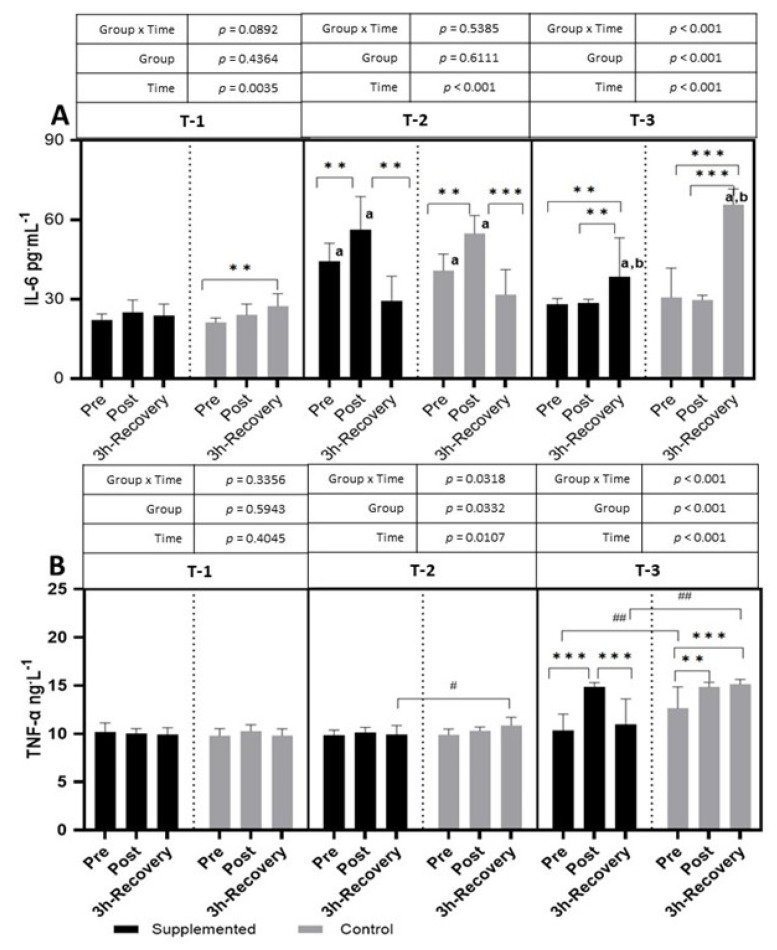
Effects of exercise and bovine colostrum supplementation on IL-6 (interleukin-6) (**A**) and TNF-α (tumor necrosis factor alpha) (**B**) in athletes. T-1: before supplementation (basal value); T-2: after 3 months of supplementation; T-3: after 6 months of supplementation. Data are means and standard deviation. ** *p* < 0.01, *** *p* < 0.001 significantly different from pre-exercise in trials; ^#^ *p* < 0.05, ^##^ *p* < 0.01, significantly different between placebo (control) and supplemented groups in trials; ^a^ significantly different from T-1; ^b^ significantly different T-2 vs. T-3.

**Figure 5 nutrients-15-04779-f005:**
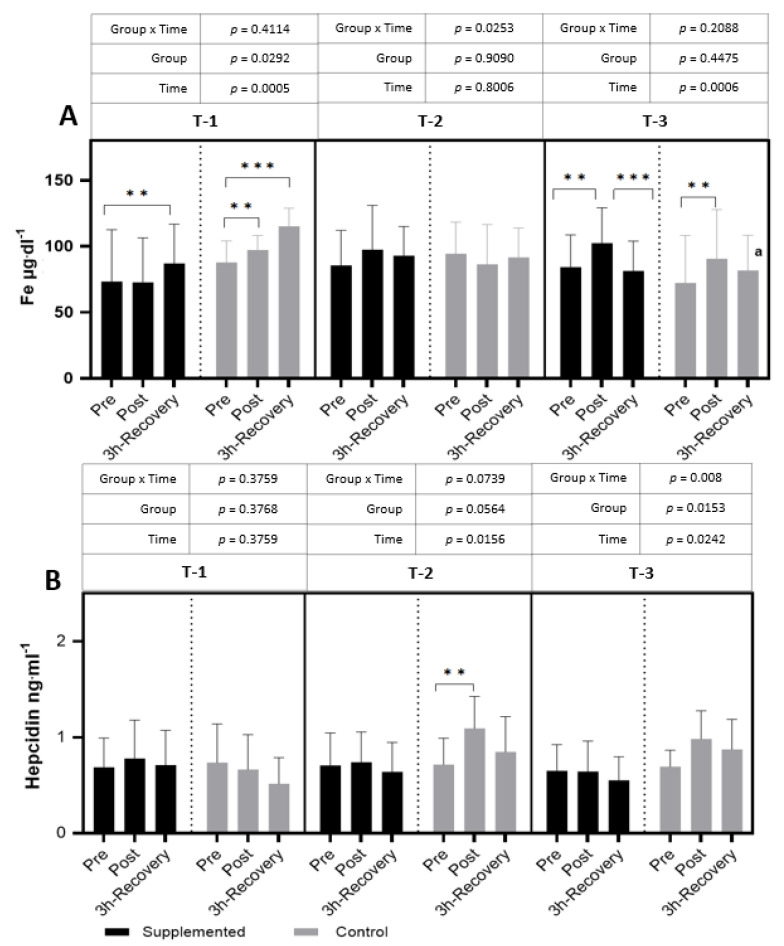
Effects of exercise and bovine colostrum supplementation on Fe (iron) (**A**) and hepcidin (**B**) in athletes. T-1: before supplementation (basal value); T-2: after 3 months of supplementation; T-3: after 6 months of supplementation. Data are means and standard deviation. ** *p* < 0.01, *** *p* < 0.001 significantly different from pre-exercise in trials; significantly different between placebo (control) and supplemented groups in trials; ^a^ significantly different from T-1.

**Figure 6 nutrients-15-04779-f006:**
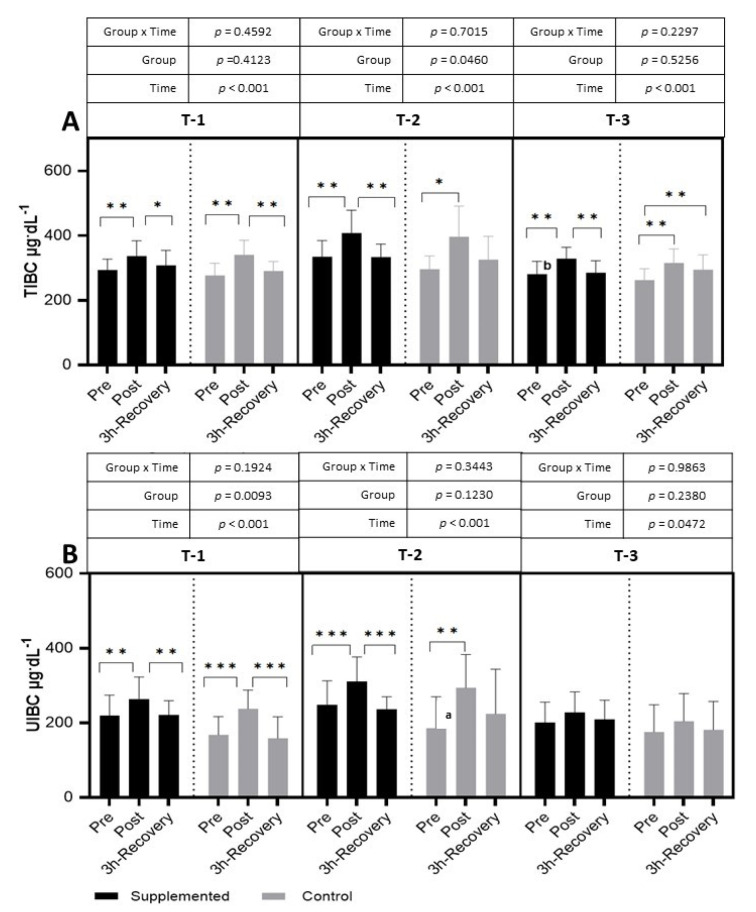
Effects of exercise and bovine colostrum supplementation on TIBC (total iron-binding capacity) (**A**) and UIBC (unsaturated iron-binding capacity) (**B**) in athletes. T-1: before supplementation (basal value); T-2: after 3 months of supplementation; T-3: after 6 months of supplementation. Data are means and standard deviation. * *p* < 0.05, ** *p* < 0.01, *** *p* < 0.001 significantly different from pre-exercise in trials; significantly different between placebo (control) and supplemented groups in trials; ^a^ significantly different from T-1; ^b^ significantly different in T-2 vs. T-3.

**Figure 7 nutrients-15-04779-f007:**
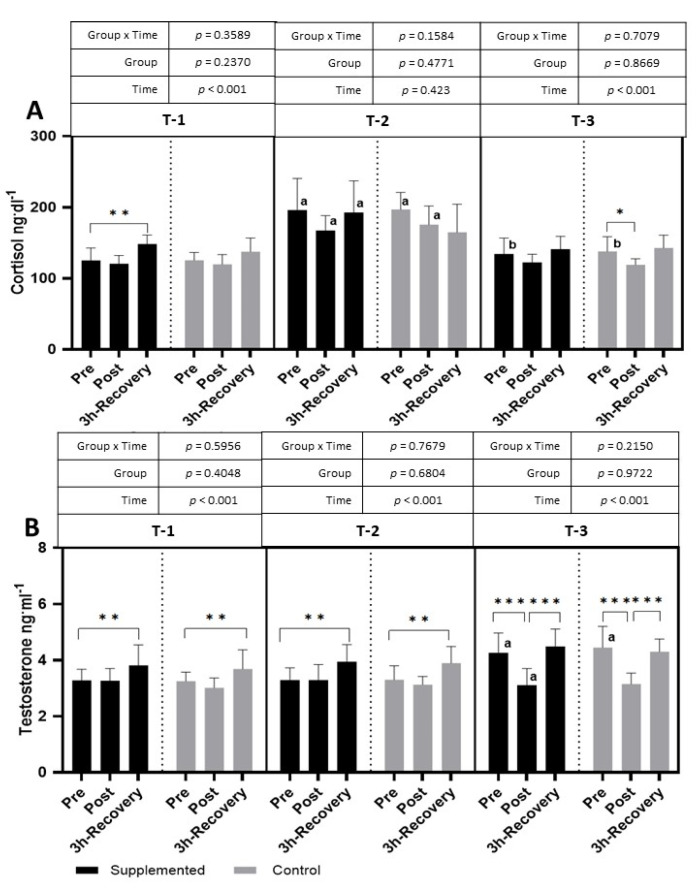
Effects of exercise and bovine colostrum supplementation on cortisol (**A**) and testosterone (**B**) in athletes. T-1: before supplementation (basal value); T-2: after 3 months of supplementation; T-3: after 6 months of supplementation. Data are means and standard deviation. * *p* < 0.05, ** *p* < 0.01, *** *p* < 0.001 significantly different from pre-exercise in trials; significantly different between placebo (control) and supplemented groups in trials; ^a^ significantly different from T-1; ^b^ significantly different in T-2 vs. T-3.

**Figure 8 nutrients-15-04779-f008:**
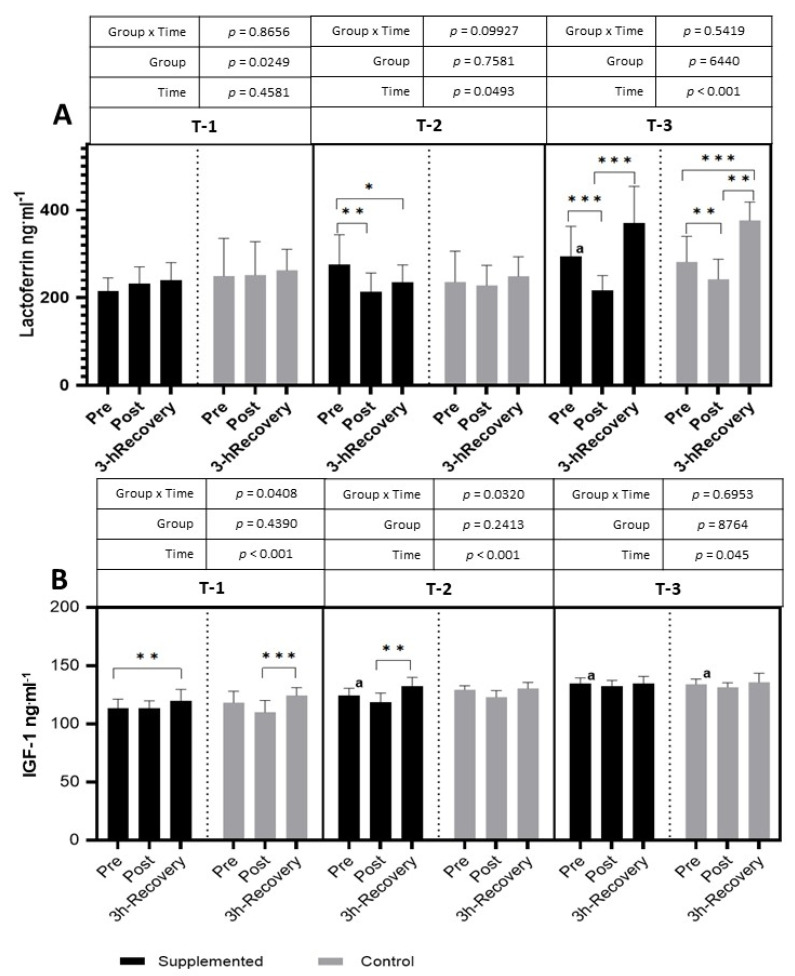
Effects of exercise and bovine colostrum supplementation on lactoferrin (**A**) and IGF-1 (insulin-like growth factor 1) (**B**) in athletes. T-1: before supplementation (basal value); T-2: after 3 months of supplementation; T-3: after 6 months of supplementation. Data are means and standard deviation. * *p* < 0.05, ** *p* < 0.01, *** *p* < 0.001 significantly different from pre-exercise in trials; significantly different between placebo (control) and supplemented groups in trials; ^a^ significantly different from T-1.

**Table 1 nutrients-15-04779-t001:** The anthropometric data of participants.

	Age (Years)	Body Mass (kg)	Body High (cm)	BMI
Supplemented (*n* = 19)	21.8 ± 5.9	73.7 ± 6.5	180 ± 5.0	22.7 ± 1.6
Placebo (*n* = 9)	19.1 ± 2.4	74.7 ± 13.8	176 ± 8.5	23.1 ± 2.3

**Table 2 nutrients-15-04779-t002:** Multistage 20 m shuttle run test (Beep Test) scores for supplemented and placebo groups before (T-1) and after the 3-month (T-2) and 6-month interventions.

	T-1		T-2		T-3	
	Level	Distance (m)	Level	Distance (m)	Level	Distance (m)
Supplemented	12.73 ± 1.29	2454.67 ± 325.49	13.33 ± 1.35	2522.67 ± 317.14	13.20 ± 1.28	2550.67 ± 308.30
Placebo	11.42 ± 0.7	2142.86 ± 186.83	11.57 ± 1.29	2160.00 ± 281.02	12.14 ± 0.98	2222.85 ± 261.74

## Data Availability

Due to ethical concerns, the datasets generated and/or analyzed during the current study that support the data cannot be made openly available; however, they are available from the corresponding author on reasonable request.

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
