# Peer review of "Long-Term Bovine Colostrum Supplementation in Football Players"

_nutrients, 2023, doi:10.3390/nu15224779_

Round 1

Reviewer 1 Report

Comments and Suggestions for Authors

Dear authors,

The submited manuscript "The long-term bovine colostrum supplementation in football 2 players" describes a well designed randomised controlled trial. A really nice work has been done in analysing and discussing the results of the study.

I just have a few minor comments regarding the manuscript that can be found in the attached file.

Comments on the Quality of English Language

English is really good in the manuscript. One minor comment about phrasing is included in the attached file.

Author Response

Thank you very much for the comments on our paper entitled „ The long-term bovine colostrum supplementation in football players”. The manuscript has been revised following all the remarks of the expert referees.

Reviewer 2 Report

Comments and Suggestions for Authors

Review

Thanks to the editor for inviting me to review the manuscript by Mirosława Cieślicka and colleagues. The manuscript aims to investigate the effects of 6 months of bovine colostrum supplementation on indicators of immune system functioning, selected parameters related to iron management, and anabolic/catabolic balance in young soccer players.

Although the article is interesting, there are some issues to be resolved.

General comments

Tables should be added in text format, not image or screenshot.

Why did no women participate in the study?

Why did the subjects ingest that amount of bovine colostrum? Is there evidence that this is an adequate dose? Would it be more appropriate to adjust the amount according to body weight?

Specific comments

Abstract

- Indicate the meaning of the acronyms of biochemical and hormonal parameters.

- Indicate the objectives of the study

Introduction

- Join the sentence of line 75 and 76.

- I recommend commenting/explaining some parameters that will be evaluated throughout the manuscript.

- Are there studies that have evaluated bovine colostrum in soccer players (10.1007/s00394-017-1401-7)? At the end of the introduction a hypothesis should be stated, as well as mentioning and describing similar studies, knowing the possible limitations that are going to be solved in this manuscript.

- Likewise, I recommend adding studies evaluating the effect of bovine colostrum on anti- and pro-inflammatory markers.

Material and Methods

2.1. Indicate the bioethics committee approval number.

- How was the sample randomized? Indicate website or instrument.

2.2. Describe the warm-up.

2.3. Were the determinations made by the authors? The instruments used, material and brand should be indicated.

2.4. Were there four doses in total or four doses in the morning and four in the evening? As I mentioned above, why did the subjects ingest that amount of bovine colostrum? Is there evidence that this is an adequate dose? Would it be more appropriate to adjust the amount according to body weight?

- Indicate the significance of the biochemical parameters.

- Why was powdered milk used as a placebo?

2.5. Why was a two-way ANOVA (group and time) not performed to evaluate the interaction between the two factors?

- I recommend calculating the effect size.

Results

- In each figure, add a letter and indicate what it represents (4 letters per figure).

- Indicate in the figure caption the meaning of the acronyms of the parameters.

- I suggest analyzing the testosterone/cortisol ratio and justifying these results. This ratio is an interesting marker to assess the efficacy of the balance between anabolic and catabolic pathways.

Harkonen N., Kuoppasalmi K., Naveri H., Tikkanen H., teen A., Adlercreutz H., Karvonen J.: Biochemical indicators in diagnosis of overstrain condition in athletes. In: Sport Med Exerc Sci, Proc Olympic Sci Congr; Eugene, USA, 1984.

Banfi, G.; Marinelli, M.; Roi, G.; Agape, V. (1993). Usefulness of Free Testosterone/Cortisol Ratio during a Season of Elite Speed Skating Athletes. International Journal of Sports Medicine, 14(7), 373-379. doi:10.1055/s-2007-1021195.

Discussion

- What would be the justification for the lower TNF alpha concentrations in the supplemented group?

- Add limitations of the study.

- Add prospective for future research.

Conclusions

-Add possible practical applications

Author Response

Response to the Reviewer 2

 Thank you very much for the comments on our paper entitled „ The long-term bovine colostrum supplementation in football players”. The manuscript has been revised following all the remarks of the expert referees.

Round 2

Reviewer 2 Report

Comments and Suggestions for Authors

The manuscript has been satisfactorily modified and I recommend its publication.